# Bias in the reporting of sex and age in biomedical research on mouse models

**Abstract** In animal-based biomedical research, both the sex and the age of the animals studied affect disease phenotypes by modifying their susceptibility, presentation and response to treatment. The accurate reporting of experimental methods and materials, including the sex and age of animals, is essential so that other researchers can build on the results of such studies. Here we use text mining to study 15,311 research papers in which mice were the focus of the study. We find that the percentage of papers reporting the sex and age of mice has increased over the past two decades: however, only about 50% of the papers published in 2014 reported these two variables. We also compared the quality of reporting in six preclinical research areas and found evidence for different levels of sex-bias in these areas: the strongest male-bias was observed in cardiovascular disease models and the strongest female-bias was found in infectious disease models. These results demonstrate the ability of text mining to contribute to the ongoing debate about the reproducibility of research, and confirm the need to continue efforts to improve the reporting of experimental methods and materials.

**OSCAR FLÓREZ-VARGAS\*, ANDY BRASS\*, GEORGE KARYSTIANIS, MICHAEL BRAMHALL, ROBERT STEVENS, SHEENA CRUICKSHANK AND GORAN NENADIC**

\*For correspondence: florezvo@ cs.manchester.ac.uk (OFV); andy. brass@manchester.ac.uk (AB)

## Introduction

Studies using animal models are important for understanding the physiopathological and therapeutic basis of human diseases. However, the translation of scientific findings from animal models to humans is far from straightforward, and more than 80% of potential therapeutics fail in human clinical trials after being successful in animal models (*Perrin, 2014*). There is a clear need, therefore, for animal research to become more reliable and reproducible (*van der Worp et al., 2010*).

The failure to translate from animal models to humans stems from various factors, and in recent years there have been growing concerns over the lack of reproducibility of results in certain areas of biomedical research (*Begley and Ellis, 2012*; *Prinz et al., 2011*; *Collins and Tabak, 2014*). One factor contributing to this observed lack of reproducibility may be inadequate reporting of experimental methods and materials (*Landis et al., 2012*; *Moher et al., 2008*; *van der Worp and Macleod, 2011*). It has been estimated that spending on preclinical research that is not reproducible amounts to $28 billion per year in the United States (*Freedman et al., 2015*).

Guidelines from the International Committee of Medical Journal Editors (ICMJE) state that the methods section of a paper "should aim to be sufficiently detailed such that others with access to the data would be able to reproduce the results" (*ICMJE, 2013*). In experiments using animals, for instance, the sex and age of the mice should be reported because they affect morphological, physiological, immunological and behavioral parameters and, hence, they influence the outcomes of experiments (*Diedrich et al., 2007*; *Wizemann and Pardue, 2001*). Sex and age are inextricably linked: it has been proposed that under natural conditions sexual selection has profound effects on the

lifespan of organisms (*Bale and Epperson, 2015*; *Maklakov and Lummaa, 2013*). Considering some taxa exceptions, the general conclusion is that in many animals (including humans), males have shorter lifespans than females (*Clutton-Brock and Isvaran, 2007*). Furthermore, from an evolutionary standpoint, these sex differences in lifespan depends to a great extent on sexually dimorphic life-history strategies (such as mating systems; *Maklakov and Lummaa, 2013*), and on genetic architecture, including both the sex chromosomes (*Nguyen and Disteche, 2006*) and the mitochondrial DNA (*Gemmell et al., 2004*).

Regarding preclinical and clinical studies, sex and age play key roles in disease phenotypes, modifying their susceptibility, presentation and response to treatment (*Arnold, 2010*). Some pathologies exhibit a clear sexual dimorphism (*Ober et al., 2008*). Using stroke as an example, it is known that its incidence is higher in men than women during their lifespan (*Mozaffarian et al., 2015*). However, recent evidence suggests that after the age of 60 years and thus post-menopause, women have more severe strokes than men (*Dehlendorff et al., 2015*). In the case of animal models, sex- and age-dependent differences in protein expression profiles were observed in the heart proteome of female and male C57BL/6 mice of two distinct age groups (14 and 100 weeks) (*Diedrich et al., 2007*). This evidence implies that sex differences must be studied across the entire lifespan in order to bring new insights into the pathogenesis of the diseases and identify targets for new drugs for both sexes and different times of life.

Guidelines, such as ARRIVE (Animal Research: Reporting In Vivo Experiments; *Kilkenny et al., 2010*), have been developed to improve the reporting of animal-based research. However, although these guidelines have been endorsed by many journals, not all papers in these journals comply with the ARRIVE guidelines (*Baker et al., 2014*). Recent years have also seen more emphasis being placed on the need for animal-based experiments to comply with the principles of the 3Rs (Replacement, Reduction and Refinement; *Burden et al., 2015*).

In this study, we have used large-scale text mining to evaluate the reporting of information about the sex and age of the mice used in a set of over 15,000 articles. In the last decade, there has been a significant amount of research in the identification of targeted biomedical information in the scientific literature via text mining (*Cohen and Hersh, 2005*; *Fleuren and Alkema, 2015*). In particular, efforts have been made to recognize protein and gene names in text (*Settles, 2005*) or other biomedical entities of interest such as electronic health records (*Meystre et al., 2008*). Our approach differs from this work in that it addresses a significantly more diverse literature space while focusing on what should be the standard information in a paper about animal-based biomedical research.

Previously we have shown that important experimental details are repeatedly omitted from papers in parasitology (*Florez-Vargas et al., 2014*) and in studies of colitis (*Bramhall et al., 2015*). Here we use syntactic rules and simple dictionary matching to extract key characteristics (such as sex and age) from papers reporting the results of experiments on mice. We investigate questions of whether the sex and age of mice are reported, the use of animals of each sex in six different areas of preclinical research, and the use of animals of each sex in four subgroups (genetics, immunology, physiopathology and therapy) for these six areas.

## Results

### System evaluation and data

We evaluated the text mining system on a set of 50 full-text articles randomly selected from our corpus of study (*Supplementary file 1*) by comparing its performance with the manual annotations of the same papers performed by two biomedical experts. The F-scores that resulted from this evaluation were around 92% for both sex and age (*Table 1*), which indicates good quality of the results (*Ananiadou et al., 2006*).

A total of 15,311 full-text articles from the PubMed Central Open Access subset as of February 2015 were processed in this study. These articles correspond to 7.15% and 27.85% of mouse experimentation articles retrieved by the same query in PubMed and PubMed Central, respectively. This corpus of documents were published between 1994 and 2014, of which 50.1% were published after 2011 (n= 7,671) (*Figure 1*). Seventy journals out of the 628 analyzed covered 30 or more articles of the corpus (*Figure 1—figure supplement 1*), which corresponds to 81.05% of papers retrieved. *PLOS ONE* contained the highest number of articles (n= 5,574; 36.41%), followed by *Journal of Experimental Medicine* (n= 931; 6.08%), and *Journal of Cell Biology* (n= 363; 2.37%).

**Table 1.** Evaluation of the performance of the text mining system.

| Characteristics | True-positives | True-negatives | False-positives | False-negatives | Precision (%) | Recall (%) | F-score (%) |
|---|---|---|---|---|---|---|---|
| Sex | 29 | 16 | 3 | 2 | 90.6 | 93.5 | 92.0 |
| Age | 31 | 14 | 1 | 4 | 96.8 | 88.5 | 92.4 |

A total of 50 articles were used as the data set to evaluate the performance of the text mining system (**Supplementary file 2D**). The precision (P), calculated as TP/(TP+FP), determines the accuracy of the system in recognizing desirable terms. The recall (R), calculated as TP/(TP+FN), produces the coverage of the system. F-score is the harmonic mean of precision and recall and it is calculated as 2*P*R/(P+R).

### Reporting of sex and age

The general and historical reporting of sex and age as experimental variables in mouse models is presented in *Figure 1*. Overall, from 1994 to 2014, about a fifth of papers did not report either the sex or the age of the mouse used in the study (*Figure 1a and 1b*). *Figure 1c* shows that the frequency of articles reporting sex and/or age in mice models has increased steadily during the last two decades, whereas missing information about these two experimental variables showed an important drop from 100% (no papers reported the sex and age of the mice in 1994 and 1995) to about 15% following a slope of approximately -0.045. Nevertheless, since 2012, the percentage of articles reporting both factors had reached only about 50% of the papers published in those years.

When the sex of the mouse model is stated in the article, experiments performed with female mice were more frequently reported than experiments performed with male mice (31.84% vs. 23.38%, Binomial test $p < 0.001$; 95% IC: 56.60 – 58.71) (*Figure 1d*). Our results showed that, historically, female mice have been reported more often than male mice, reaching a plateau of about 33% since the last decade (2004 – 2014) (*Figure 1e*). In addition, the use of both sexes in mice experiments stratified by sex showed the lowest improvement over time (*Figure 1e*); with a maximum of about 10% of the articles since 2006. Reporting of mouse age improved steadily from 1999 to 2006 (*Figure 1f*), at which point age is reported more than 50% of the time; since 2010 age reporting has plateaued, with between 65 and 70% of articles each year mentioning the age of mice.

In order to identify whether there are general features common on reporting sex and age as experimental variables to any biomedical field, we assessed six main preclinical research topics as defined by their impact on human health (*WHO, 2014*), including: cardiovascular diseases; cancer; diabetes mellitus; lung diseases; infectious diseases; and neurological disorders. A two-way ANOVA without replication was performed to assess the difference in reporting sex and age for each field. Our results showed statistically significant differences, $p < 0.05$, indicating that the reporting of these experimental factors varies across biomedical fields (*Figure 2*). In identifying the sex and age of the mouse, for instance, studies on diabetes showed the highest frequency (68%), whereas studies on cancer showed the lowest frequency (48%) (*Figure 2a*). Studies on cancer reported the worst results regarding missing information about sex (33%) or age (37%) of the mice used (*Figure 2b and 2c*). Overall, the best results in reporting sex and age were achieved by the studies on neurological disorders (*Figure 2a, 2b and 2c*).

For a more detailed analysis of sex-based reporting, the six groups of diseases were divided into four subgroups according to the characterization of the disease models via genetics, immunology, physiopathology and therapy. Our results suggest that there is a preference for studying the immunology of these diseases by using female mouse models, whereas there is a tendency to use male mouse models for studying their genetic basis (*Figure 3a and 3b*). Both in physiopathology and in therapy subgroups, male mice were more frequently studied in models of cardiovascular diseases, diabetes and neurological disorders, and female mice in models of cancer, lung diseases and infectious diseases (*Figure 3c and 3d*).

In order to further test whether the observations about the reporting of sex in the experimental mouse models were conserved even in specific cases, we focused the analysis on one particular disease per group as follows: myocardial ischemia (cardiovascular disease); diabetes mellitus type 2 (diabetes); chronic obstructive pulmonary disease (lung disease); Alzheimer's (neurological disorder). Three diseases were

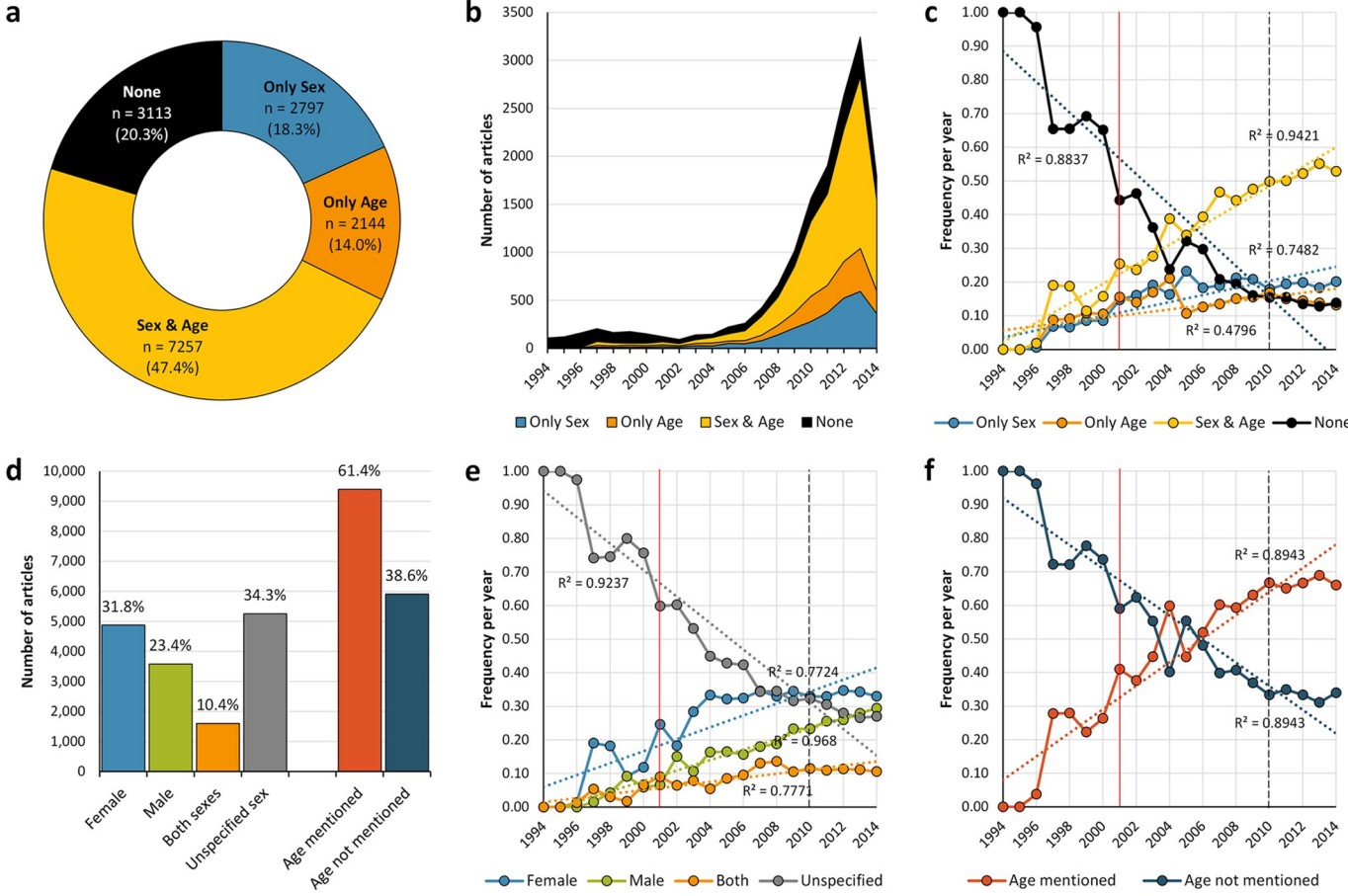

**Figure 1.** General distribution and historical change of reporting and non-reporting of sex and/or age in mouse-model experiments. Pie-chart (**a**) showing an overview of the reporting and non-reporting (none) of sex only, age, or both sex and age in a set of 15,311 studies published between 1994 and 2014 by stating the number and percentage of articles in each portion. The chronological change of the reporting and non-reporting is displayed both in a stacked area plot (**b**) and a scatter plot after normalization [per articles/year] (**c**). The chronological changes show that most of the articles assessed were published during the last decade (**b**), and that the improvement of reporting of these two biological factors started before, and not after, the US Institute of Medicine report in 2001 (*Wizemann and Pardue, 2001*) [indicated with a vertical red line] or the introduction of ARRIVE guideline (*Kilkenny et al., 2010*) [indicated with a vertical black dashed line] (**c**). Bar-chart (**d**) showing the number and percentage of articles reporting/ not reporting of sex by sex [females only, males only, or both sexes either by mixing or separating them] or age. The chronological change of the reporting and non-reporting of sex by sex (**e**), and age (**f**), is displayed in scatter plots after normalization [per articles/year].

The following source data and figure supplement are available for figure 1:

**Source data 1.** PubMed search terms used for each disease group and their approaches.

**Source data 2.** Example rules for identification of sex and age.

**Figure supplement 1.** Reporting of sex or age in mouse-model experiments by journal.

included in the case of infectious diseases that are among the most frequently reported causes of death world-wide (*WHO, 2014*), *i.e.* tuberculosis, HIV and malaria. Melanoma was included for the cancer group since it is a highly aggressive and notoriously chemoresistant form of cancer; making it a widely used tumor model (*Herlyn and Fukunaga-Kalabis, 2010*). Overall, our results suggest that in most cases there is a similar pattern of reporting as that found for the biomedical fields assessed to which these diseases belong (*Figure 4*).

Bibliometric parameters were used to determine if they were associated with the quality of method reporting. We used as journal metrics both the journal impact factor from the Institute for Scientific Information (ISI) Web of Knowledge's Journal Citation Report (2014), and h-

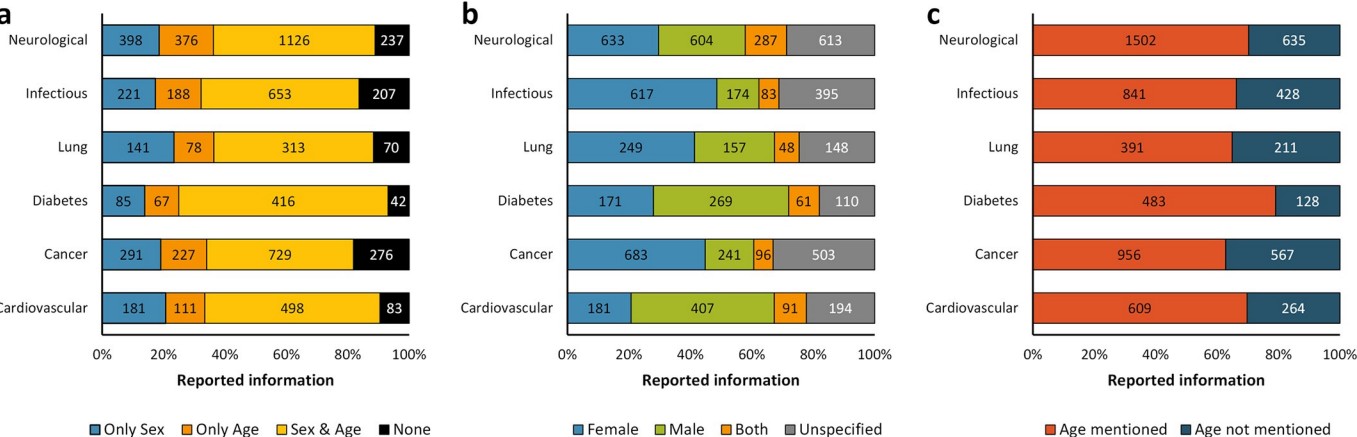

**Figure 2.** Distribution of reporting of the sex and age in mouse model of a group of diseases. The reporting of these variables was assessed for six groups of diseases from the top 10 causes of death according to the W.H.O. This analysis was performed in the set of 14,225 articles published from 2001, when the US Institute of Medicine report was published (*Wizemann and Pardue, 2001*) and when the non-reporting of sex and age together dropped about 50% –avoiding misinterpretations [*Figure 1c*], to 2014. The distribution is presented in stacked bar charts that illustrate the percentage of the reporting and non-reporting for both biological variables overall (**a**) and discriminated by variable: sex (**b**) and age (**c**); stating the number of articles corresponding to each percentage inside the stacks. A two-way ANOVA without replication was performed to assess the difference in reporting of the sex [p = 0.005] and age [p = 0.028] for each disease, indicating that the reporting and non-reporting of these biological factors varies across these diseases.

index from the SCImago Journal and Country Rank (2014). No correlation was observed between the reporting of sex or age as experimental variables and the journal impact factor and h-index of the 70 journals that covered 30 or more articles of the corpus (*Figure 5*).

## Discussion

Whilst this work constitutes the largest analysis of the reporting of sex and age data for mouse-based research to date, our sample does not represent the entire biomedical literature: not all journals are found in the PubMed Central Open Access subset, and some of the journals that deposit their contents into PubMed Central include only some of their articles in the Open Access subset. This is undoubtedly a limitation of our study.

We selected the mouse as a model because it is probably the most comprehensive and well-characterized model in the life sciences. Researchers rely on mouse models to mimic human disease conditions for several reasons. One of the main reasons is that mouse and human genomes are genetically similar – about 90% of human genes have direct orthologues with mice (*Yue et al., 2014*). Moreover, as animal models, mice are convenient due to their small size, short lifespan (up to two years), and quick generation time; three weeks for gestation and from 6 to 8 weeks to reach sexual maturity.

Therefore, they can be easily housed and maintained, can be genetically manipulated to define gene function in a whole body system and a large number of mice can be studied in a relatively short period of time. This, for instance, allows scientists to study cell/cell interactions in the tissue environment and thus cause and effect relationships in a controlled situation.

Despite the implications for interpretation and reproducibility of experimental findings, the sex and age of the experimental subjects are often not recorded in scientific reports (*Kilkenny et al., 2009*). In agreement with previous reports, the evidence presented in this study showed that the lack of reporting of key methodological parameters in mouse experiments is still a cause of concern; only about half of the papers published in 2014 stated both sex and age of the mice as experimental variables (*Figure 1c*). The reason why these variables are not described is unclear, since this simple information is always available to researchers. We do not believe it is a space issue, because it is possible to describe them, including mice number and mouse strain, in about 40 characters of text (*e.g.* ten C57BL/6 female mice (6-8-weeks old)). Whilst an improvement in the reporting of mouse sex and age has been observed over time, this is not solely attributable to the introduction of journal guidelines, because this improvement started before the publication of

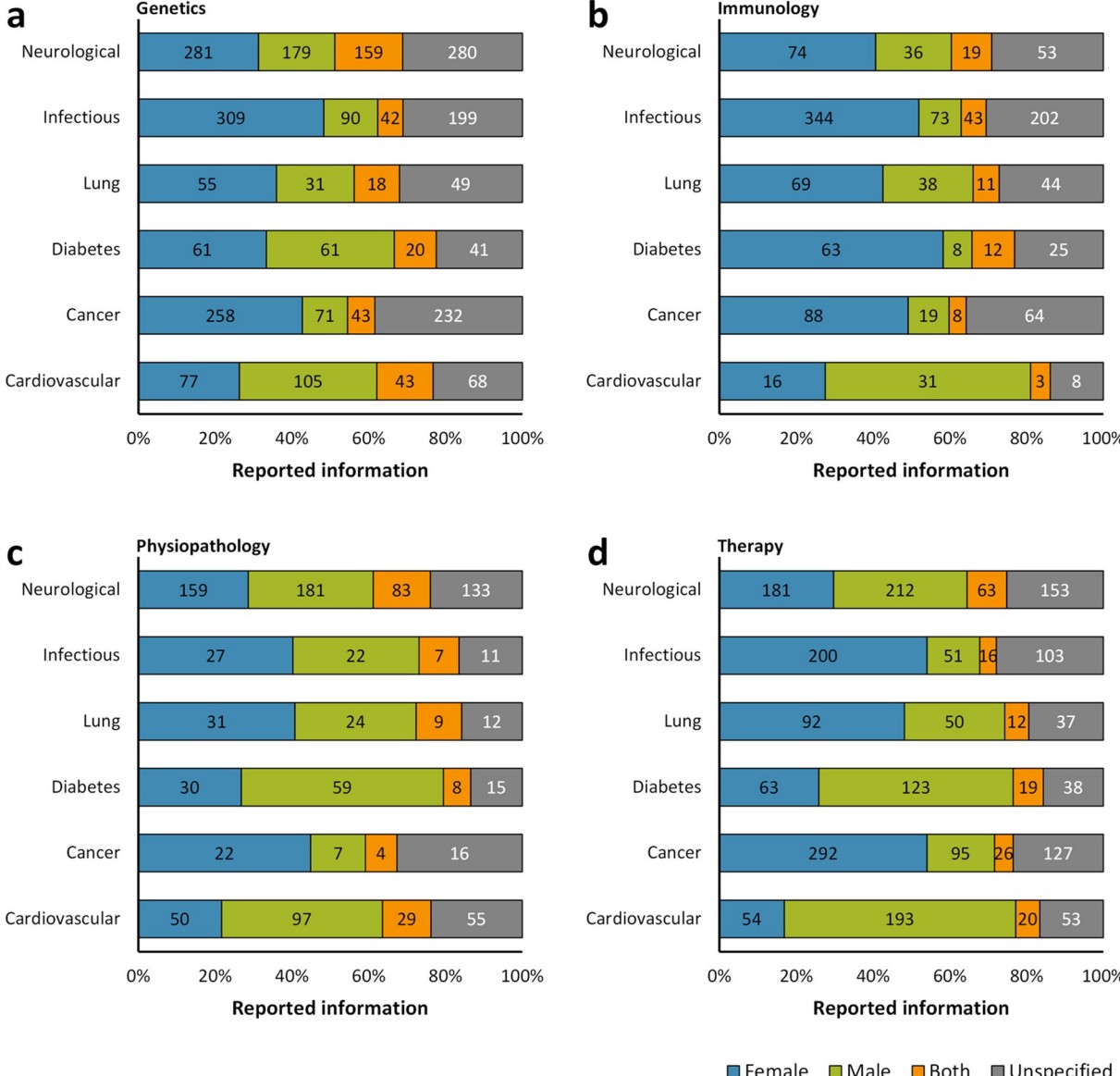

**Figure 3.** Distribution of reporting of the sex in mouse model of a group of diseases by research approach. The reporting of sex was assessed for each disease by the topic of research whether genetics (**a**), immunology (**b**), physiopathology (**c**), or therapy (**d**). This analysis was performed in the set of 14,225 articles published from 2001, when the US Institute of Medicine report was published (*Wizemann and Pardue, 2001*) [*Figure 1c*], to 2014. The distribution is presented in stacked bar charts that illustrate the percentage of the reporting and non-reporting for the sex; stating the number of articles corresponding to each percentage inside the stacks. A two-way ANOVA without replication was performed to assess the difference in reporting of the sex for genetics [p = 0.0009], immunology [p = 0.0074], physiopathology [p < 0.0001], and therapy [p = 0.1165], indicating that the reporting and non-reporting of these biological factors varies across most of these biomedical approaches.

the ARRIVE guidelines in 2010 (*Kilkenny et al., 2010*). In fact, a follow-up study in 2012 showed that while sex and age reporting had improved post-ARRIVE, journals that enforced the ARRIVE guidelines as a condition of publication still failed to publish sex and age in all cases (*Baker et al., 2014*). The observed improvements may also therefore be a result of a growing recognition of the importance of sex and age as experimental factors that may affect study outcomes.

An analysis of the scientific literature leads to the general conclusion that the males in both human and other animals are studied much more than their female counterparts. This conclusion is based mainly on the results of two studies that manually surveyed a set of biomedical articles (*Beery and Zucker, 2011*;

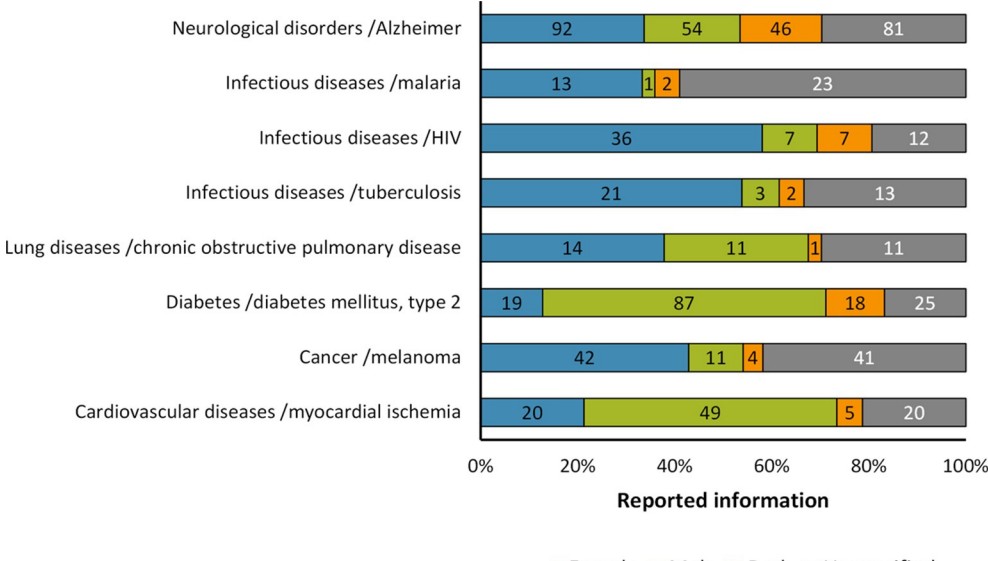

**Figure 4.** Distribution of reporting of the sex in mouse model of diseases. The graph shows the reporting in particular diseases. All these diseases that are among the most frequently reported causes of death world-wide or commonly used models. The distribution is presented in stacked bar charts that illustrate the percentage of the reporting and non-reporting for the sex; stating the number of articles corresponding to each percentage inside the stacks. This analysis was performed in a set of 791 articles; see *Figure 1—source data 1*.

*Taylor et al., 2011*). However, our results showed otherwise in mouse-based models: 31.84% and 23.38% of all papers assessed were on studies performed on female and male mice, respectively (*Figure 1d*). This could be explained by some practical advantages of using female rather than male mice: they are cheaper to house; they are less aggressive to each other and to experimenters; and they are smaller, requiring less weight-administered drug. In addition, the apparent contradiction between this observation and the previous reports might be related to the sample size and study design; our sample size was larger and we surveyed a much broader range of disciplines. In addition, although rodent models (that is, mouse and rat models) accounted for, respectively, 50% and 80% of the papers considered in these other studies (*Beery and Zucker, 2011*; *Taylor et al., 2011*), they also included other species, such as cat, dog and monkey models: moreover, these two studies did not explore sex bias by species.

In preclinical studies, furthermore, we noted an important sex- and age-bias in mouse-based disease models (*Figure 2b and 2c*). Among the main preclinical research topics assessed, we observed the strongest male-bias in cardiovascular disease models (2.25:1) and the strongest female-bias in infectious disease models (3.54:1)

(*Figure 2b*). This situation still persists: between 2012 and 2014, about 70% and 77% of research articles assessed on these two disease models are still biased towards male and female mice, respectively. These pathologies and many others, exhibit important sexual dimorphisms, which are not only inherent to genetic differences, but also to hormonal influence (*Case et al., 2013*; *Gilks et al., 2014*). For example, in the study of hypertension, one of the major risk factors for cardiovascular disease, a greater increase in blood pressure was reported in gonad-intact XY males than XX females using the four core genotype in the MF1 mouse model. However, the mean arterial pressure was greater in gonadectomized XX mice compared with XY mice regardless of whether the mice were born with testes or with ovaries (*Ji et al., 2010*). On the other hand, in the case of infectious diseases, females have a more robust immune system than males – both the innate and adaptive immune responses, which makes them less susceptible to developing many infections (mainly Th1-type infections), although it increases the risk of developing autoimmune diseases due to their trend to develop a stronger pro-inflammatory response (*Pennell et al., 2012*). Interestingly, we also observed that the sex-bias could change in a particular disease

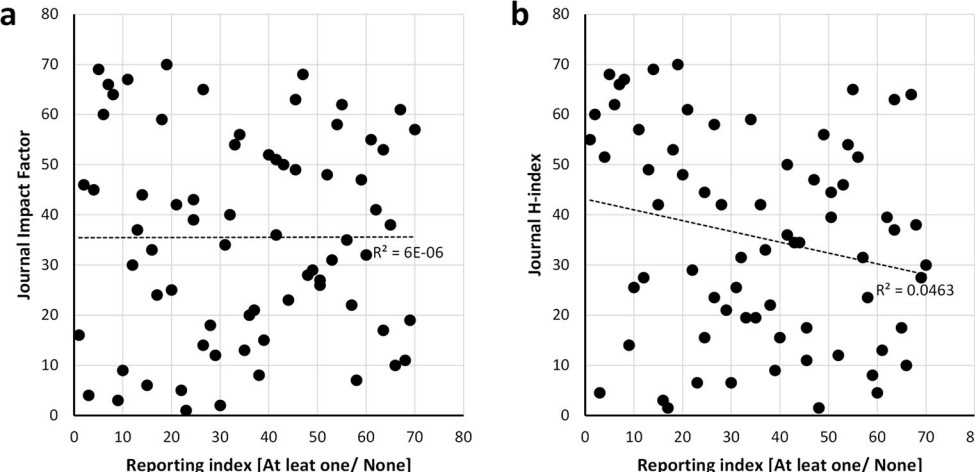

**Figure 5.** Scatter plots showing the relationship between the reporting and the bibliometric indices. Journal impact factor in which the papers were published (**a**) and h-index of journals (**b**). Spearman's rank correlation coefficient r square is shown alongside the regression lines. The scatter plots show that there is no correlation between the reporting and impact factor [r =0.002, p = 0.984] data from the Journal Citation Report (year 2014) and journal h-index [r =-0.215, p = 0.073] data from the SCImago Journal and Country Rank (year 2014). Analysis conducted on the 70 journals that published 30 or more articles of the 15,311 studies returned by searching the PubMed Central Open Access subset as of February 2015.

mouse model according to the biomedical study. Diabetes disease mouse models exemplified this situation. From a global point of view, this disease was found to be male-biased (1.57:1) (*Figure 2b*). However, in studies related with the immunology of diabetes, there was a strong female-bias (7.87:1) (*Figure 3b*); a change that remains in the study of diabetes mellitus type 2 (*Figure 4*).

In order to balance sex of animals and cells in preclinical studies, the National Institutes of Health (NIH) have proposed a multi-dimensional initiative, which includes, among other things, extramural training on experimental design and data analysis by sex (*Clayton and Collins, 2014*). Regarding this initiative, new ideas have been proposed to achieve, and sustain, the sex balance in biomedical research (*McCullough et al., 2014*). In this context, our study provides an implementation of text mining to assess reporting of experimental factors. By knowing where there is imbalance for a particular variable, it is possible to address it in a cost-effective manner. This not only directly contributes to the comparability of experimental work, but also to the reproducibility of findings. To address this problem some journals are already introducing editorial measures and methods checklists in order to improve the quality of scientific reporting (*Nature, 2013*). Nevertheless, whilst journal checklists may make reference to

species strain, sex and age of animals, the main focus is on statistical issues to ensure repeatability, rather than biological factors that modify the outcomes.

We hope that our text mining strategy can be used to explore other aspects of how experimental methods and materials are reported in the literature, and thus contribute to efforts to improve the reproducibility of biomedical research. It might also be possible to employ text mining to pre-screen manuscripts when they are submitted to journals.

## Methods

### Search strategy and data

A literature search was carried out in Medline via PubMed in order to identify research articles that deal with mouse experimentation. The database was searched in March 2015 for articles that were published between 1st January, 1994 and 31st December, 2014 using the terms as they appear in *Figure 1—source data 1*. To ensure maximum specificity in the search, searching was limited to articles where the MeSH (Medical Subject Headings) "Mouse" term indicated the major focus of the article; moreover the keywords "Mouse" or "Mice" had to be stated in the title. This also prevented articles that made only passing references to mouse work from entering the dataset and

ensured a high quality corpus for analysis. The search was restricted to English language papers and to research articles (excluding review articles). In addition, to obtain full text articles, we restricted the PubMed search to include only those in PubMed Central by adding the special term "pubmed pmc[sb]" in the query. The PubMed Identifiers (PMID) were then converted to the respective PubMed Central (PMC) reference numbers which were acquired by querying the PubMed Central Open Access subset as of February 2015, which contains over one million full-text articles to date.

In order to assess particular areas in which there is strong scientific interest world-wide, we analyzed experiments performed in mouse models for six groups of diseases from the top 10 causes of death according to the W.H.O. in high, low and middle income countries (*WHO, 2014*). The six disease groups were as follows: cardiovascular diseases; cancer; diabetes mellitus; lung diseases; infectious diseases; and neurological disorders. Some causes of death did not apply for our study, *e.g.* road injury. HIV/AIDS, tuberculosis and other infections, for instance, were included in the infectious diseases group. A group for cancer was created in a similar way. An example disease for each of the six disease groups was also included. In addition, as there are different approaches to assess disease models according to the research field, *e.g.* immunology, genetics etc., each of these areas were divided into a series of subgroups by using the Subheading MeSH terms "genetics", "immunology", "physiopathology" and "therapy" (*Figure 1—source data 1*). These four approaches were chosen because of their importance for understanding the molecular and physiological basis of diseases, as well as for developing novel therapeutic agents for their treatment. These subjects were used to find if these disease models are being assessed consistently by sex and age.

In 2001 the US Institute of Medicine report (*Wizemann and Pardue, 2001*) concluded that sex matters in diseases and response to therapy; we therefore decided to explore any changes before and after the report by selecting articles between 1994 and 2014. This time span allows us to assess the impact of this report on the reporting of this experimental factor. In order to avoid misinterpretation due to low number of papers prior to 2001, the analysis for groups and subgroups was applied to articles published after 1st January 2001.

### Sex and Age identification: data sets

The text mining approach involved the design and implementation of generic rule-based patterns, which identify age and sex mentions in text. The rules were based on lexical patterns engineered from a sample of 40 full-text articles manually selected from PubMed through a thematic query of interest as follows: "Mice"[Mesh] AND (mouse[ti] OR mice[ti]) AND "animals"[MeSH Terms:noexp] AND Journal Article [ptyp] AND English[lang]. The first 40 papers that mentioned the sex and/or age of the mice were selected (*Supplementary file 2A*).

The *age* rules were based on lexical patterns mentioning age clues, *e.g.* "*aged 3 to 8 weeks old*". Similarly, the *sex* rules were designed around word matching aiming to identify male, female or both sexes in mice, *e.g.* "*mice of either sex were used*".

The rules were created and applied via GATE (*Cunningham et al., 2013*) for Windows version 8.1; an open source free software enabling the design and implementation of information extraction systems in unstructured text with the crafted rules following its notation (https://gate.ac.uk/). The number of crafted rules was 12 for sex and 18 for age. *Figure 1—source data 2* presents examples of rules for both the sex and age whereas *Supplementary file 3* displays all the utilized rules for the two characteristics.

The results generated by text mining were then integrated at the document level. In cases where several different candidate mentions for a single characteristic, *i.e.* sex or age, are recognized in a given document, we 'unified' them to get document level annotations using the following approach: if multiple mentions of different lengths occur, the longest is selected (usually the most informative) aiming to have one mention for both the sex and age per document, and where mentions are of the same length, the first one is chosen.

Since our method focuses on the recognition of age and sex at the mention level per document, we hypothesize that it is highly unlikely for researchers to report key information about animal models that they did not use. In order to further support this hypothesis, 40 full-text articles were randomly selected from our corpus and through manual inspection, we concluded that indeed, if there are mentions in text (particularly in the Method section) of specific age and sex (together) these are attributed to the mice used in the animal experiments and no further mentions were reported (*Supplementary file 2B*).

The randomness was modelled by using the "=RANDBETWEEN()" function in Microsoft Office Excel for Windows version 2013 as follows: according to the text mining results, each paper of the corpus of articles with a positive mention of the sex and/or age of the mice was assign a random number from 1 to 40. The first 40 papers identified with the random number 1 were selected.

Finally, to further enhance the performance of the rules, we applied this strategy to a development set of 70 full-text documents (*Supplementary file 3C*). These articles were randomly selected from our corpus by using the "=RANDBETWEEN()" function in Microsoft Office Excel for Windows version 2013; assigning to each paper a random number from 1 to 5. After sorting by the "Year" column, the first five papers identified with the random number 1 were selected by each year group. The mentions of age and sex in both corpus were manually identified and reviewed by the first author, who has a background in the field of biomedical research. A summary of the data sets used in this study is presented in *Table 2*.

### System evaluation

The performance of the text mining system was evaluated at the document level by considering whether the returned mentions were correctly the sex and age of the mice studied. In order to create an evaluation dataset, 50 full-text articles were randomly selected from our corpus of study (*Supplementary file 3D*) and were manually double-annotated for both the age and the sex by the first and fourth authors due to their biomedical expertise. There was no disagreement between the manual annotations performed by two biomedical experts. The

randomness was modelled by using the "=RANDBETWEEN()" function in Microsoft Office Excel for Windows version 2013 as follows: a random number from 1 to 50 was assigned to each paper. The first 50 papers identified with the random number 1 were selected.

Precision (P), Recall (R) and F-score were calculated for both the age and the sex using the standard metrics (*Ananiadou et al., 2006*; *Hotho et al., 2005*), which rely on the number of true- and false-positive (TP and FP), and true- and false-negative (TN and FN) cases. The precision (P), calculated as TP/(TP+FP), determines the accuracy of the system in recognizing desirable terms. The recall (R), calculated as TP/(TP+FN), produces the coverage of the system. Often, there is an inverse relationship between precision and recall; when an increase occurs in precision, a simultaneous decrease is observed in recall and vice versa. Therefore, the F-score was also used for evaluating the performance of information extraction systems due to its harmonic mean of precision and recall and it is calculated as 2*P*R/(P+R). *Table 1* shows the results of the evaluation set at the document level.

Despite the overall positive performance of our text mining system, there were some results that lead to false-positive and false-negative results due to the relatively complex expressions. False-negative results regarding age mentions occurred because the rules are based on syntactical patterns that require a numeric range between specific time units, *i.e.*, days, weeks and months. For example, in the sentence *"Nineteen animals, including males and females, of ages from postnatal day (P) 7 to several months were deeply anesthetized by isoflurane and decapitated"* (*Arbogast et al., 2013*), age

**Table 2.** Summary of the data sets used in this study.

| Sets of articles | Number of articles | Task | File |
|---|---|---|---|
| Data 1 | 15,311 | Corpus for assessing reporting of the sex and age of the mice | *Supplementary file 1** |
| Data 2 | 40 | Creating the text-mining rules | *Supplementary file 2A* |
| Data 3 | 40 | Manual inspection for finding the location of the mention of the sex and age of the mice | *Supplementary file 2B* |
| Data 4 | 70 | Enhancing the performance of the text-mining rules | *Supplementary file 2C* |
| Data 5 | 50 | Evaluating the text-mining system | *Supplementary file 2D* |

***Supplementary file 1** also contains data sets of the six groups of diseases analyzed (cardiovascular diseases; cancer; diabetes mellitus; lung diseases; infectious diseases; and neurological disorders), as well as of the different approaches to assess the disease models (*i.e.* genetics, immunology, physiopathology and therapy), and the disease example for each of the six disease groups.

is not mentioned as a range concept of days (or weeks or months) but as *"postnatal days to several months"* without indicating the exact number of months. Cases like this suggest that an extension of the current rule set could lead to an improvement towards the system's performance. False-negative results regarding sex mentions occurred because the rules for the sex recognition is rather straightforward with a simple dictionary matching (minimal), which, as a consequence, does not enable the identification of the sex through inference, *e.g.* when sex-specific proxy elements are mentioned, such as pregnancy. For example, in the sentence *"Primary mouse mammary epithelial (PMME) cells were isolated from 15-d timed-pregnant CD-1 mice"* (*Lin et al., 1995*) are expected to be missed since the sex of the mice used in this experiment is female and is being inferred by the word *"pregnant"*.

On the other hand, the application of a dictionary approach generated interestingly few false-positives in the sex recognition. This is because the system identified words like *"male"* or *"female"* early in text, whereas in the actual experiment the scientists did not report any specific sex for the selected model. For example, in the sentence *"The colony of animals carrying the Pak1ip1mray allele is maintained by crossing male carriers with FVB/NJ females. All embryos presented in the phenotypic analysis of this study were produced from carriers crossed for at least four generations onto an FVB/NJ background"* (*Ross et al., 2013*), the sex of the embryos was not established even though the findings relied on them. Other cases were: *"Epithelial cells were derived from tracheas of 3-weeks old Gprc5a mice"* and *"by peritoneal into 8–12 weeks old C56Bl/6 mice"*. Cases like these suggest that the implementation of a more sophisticated system that could target common syntactical patterns observed in text (similar to those for the characteristic of age) will contribute to an improvement of the precision and performance of the system. This could explain why sex had the lower precision (90.6%) of the two analyzed factors (*Table 1*). On the contrary, there was only one false positive (referring to the embryonic stage of the mice) although the real age could not be recognized directly due to not being explicitly expressed; *"Genomic DNA and pooled total RNAs were isolated from CRL2196 cells and from various tissues, ages and lineages of mice as indicated, using standard methods and Trizol (Invitrogen),*

*respectively"* (*Li et al., 2014*). The more refined rules led to an increased precision of 96.8% (*Table 1*).

Although our text mining protocol does produce reliable results, the returned results are merely an indication of how text mining can be used to improve issues such as the under-reporting of key information in mouse based studies. There is room to improve the applied text mining strategy. Crafting more flexible rules for the capture of age and including more specific ones for the recognition of sex could improve the generated results and reveal a clearer picture of the reporting of these variables in the biomedical field. While the variety of the observed common lexical patterns was not wide in the training and development sets (*Supplementary files 2A and 2C*), a larger set could reveal other patterns that could help increase the recall. Nevertheless, the F-measure of 92% (*Table 1*) gives enough confidence in using this automated method to assess the incidence of reporting sex and age in biomedical articles.

### Statistical analysis

The frequencies of reporting of sex and age by articles were determined in Microsoft Office Excel 2013 for Windows. Differences in reporting of sex and age of mice in multiple models of diseases, as well as the use of each sex by the topic of research for each disease were assessed by two-way ANOVA without replication. An index of the reporting for each journal was calculated by dividing the number of articles that report the sex and/or age of the mouse by the number of articles that do not report any of these biological variables. Spearman's rank correlations were calculated between the reporting index and impact factor from the Journal Citation Report, and h-index journal from the SCImago Journal and Country Rank. All statistical analysis was performed by using the GraphPad Prism software for Windows version 6.05, La Jolla CA, (www.graphpad.com). Graphical representation of the data was performed using Microsoft Office Excel for Windows version 2013.

### Acknowledgement
The authors thank Dr Geraint Duck for his valuable suggestions into making this method automatic.

**Oscar Flórez-Vargas** is in the Bio-health Informatics Group, School of Computer Science, The University of

Manchester, Manchester, United Kingdom

**Andy Brass** is in the Bio-health Informatics Group, School of Computer Science, The University of Manchester, Manchester, United Kingdom

**George Karystianis** is in the Text Mining Group, School of Computer Science, The University of Manchester, Manchester, United Kingdom

**Michael Bramhall** is in the Bio-health Informatics Group, School of Computer Science, The University of Manchester, Manchester, United Kingdom

http://orcid.org/0000-0001-5938-158X

**Robert Stevens** is in the Bio-health Informatics Group, School of Computer Science, The University of Manchester, Manchester, United Kingdom

**Sheena Cruickshank** is in the Manchester Immunology Group, Faculty of Life Science, The University of Manchester, Manchester, United Kingdom

**Goran Nenadic** is in the Text Mining Group, School of Computer Science, The University of Manchester, Manchester, United Kingdom, and the Manchester Institute of Biotechnology, University of Manchester, Manchester, United Kingdom

## Author contributions

OF-V, Conception and design, Acquisition of data, Analysis and interpretation of data, Drafting or revising the article; AB, RS, GN, Conception and design, Analysis and interpretation of data, Drafting or revising the article; GK, Acquisition of data, Analysis and interpretation of data, Drafting or revising the article; MB, SC, Analysis and interpretation of data, Drafting or revising the article

**Competing interests:** The authors declare that no competing interests exist.

## Additional files

### Supplementary files

• Supplementary file 1. Corpus for assessing reporting of the sex and age of the mice.

• Supplementary file 2. (A) Set of articles for creating the text-mining rules. (B) Set of articles for finding the location of the mention of the sex and age of the mice. (C) Set of articles for enhancing the performance of the text-mining rules. (D) Set of articles for evaluating the text-mining system.

• Supplementary file 3. Rules used to identify the sex and age of experimental mouse models.

## Funding

| Funder | Grant reference number | Author |
|---|---|---|
| Departamento Administrativo de Ciencia, Tecnología e Innovación | Scholarship | Oscar Flórez-Vargas |
| Engineering and Physical Sciences Research Council | Scholarship | Michael Bramhall |
| Epistem Ltd | Scholarship | Michael Bramhall |

The funders had no role in study design, data collection and interpretation, or the decision to submit the work for publication.

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
