## [Decision Letter]

Thank you for submitting your article "Bias in the reporting of sex and age as experimental factors across biomedical research on mouse models" to *eLife* for consideration as a Feature Article. Your article has been reviewed by two peer reviewers, and the evaluation has been overseen by a guest Reviewing Editor (Chris Mungall), a Senior Editor (Fiona Watt) and the Features Editor (Peter Rodgers).

The following individuals involved in review of your submission have agreed to reveal their identity: Terry Meehan (peer reviewer) and Nicole Vasilevsky (peer reviewer).

The reviewers have discussed the reviews with one another and the Reviewing Editor has drafted this decision to help you prepare a revised submission.

The Features Editor will also contact you separately about some editorial issues that you will need to address.

Summary:

The authors have developed a text mining approach to assess bias in the reporting of sex and age of mouse model experiments reported in the scientific literature over the last two decades. The manuscript provides sufficient background as to why reporting this information is important. The methods and results are sound. The authors find while there is general improvement in reporting these variables, about half of the papers published in 2014 still lack these variables. An interesting gender-bias is noted in different fields of study.

Quantifying the problem will, it is hoped, drive journals and authors to include this information in the biomedical literature. This work could also be used by journals to help standardize descriptions of age and sex in the literature. It would also be good if the authors could rerun this analysis five years from now to see if the situation has changed.

Before publication, the following comments should be addressed.

Essential revisions:

1) In this analysis, you examined approximately 600 journals and 70 of those journals covered 30 or more articles of the corpus. In those 70 journals, do the instructions to authors require or request that sex and age are reported in the manuscripts? Are the authors complying with the journal reporting requirements?

2) I like how you show that there is an increase in the reporting of sex and age over time. In the Discussion, can you speculate as to why this is? Did journal reporting requirements get stricter, or did the ARRIVE guidelines come out at the time point when we started to see an increase?

3) Please explain why text mined articles were restricted to those with "Mice/Mouse" in the title. This eliminates many research articles. (For example, Nature is included in Figure 1—figure supplement 1, but Science is not: I suspect this is because of editorial differences in title rules, which may bias conclusions from this study). There is probably a compelling reason (e.g. automated sex and age detection breaks down when humans and mice data are described in the same article) but please tell the reader why, and note this as a limitation in the discussion.

4) In the Discussion, conjecture on why there is gender bias for females in many mouse studies. There are a number of references on mouse housing density – is gender assessed in these (e.g. some strains of male mice fight each other when caged, necessitating separate cages and driving up costs)?

---

## [Author Response]

*Essential revisions: 1) In this analysis, you examined approximately 600 journals and 70 of those journals covered 30 or more articles of the corpus. In those 70 journals, do the instructions to authors require or request that sex and age are reported in the manuscripts? Are the authors complying with the journal reporting requirements?*

We checked this and found that 50 out of the 70 journals requested the reporting of sex and age via the ARRIVE guidelines, and only one (Diabetologia) requires the reporting of these factors in the author’s guidelines. Accordingly, we have modified Supplementary Figure 1 in order to show this information.

*2) I like how you show that there is an increase in the reporting of sex and age over time. In the Discussion, can you speculate as to why this is? Did journal reporting requirements get stricter, or did the ARRIVE guidelines come out at the time point when we started to see an increase?*

We appreciate your comment, and we have addressed this point in the second paragraph in the Discussion section. In addition, we have also indicated in Figure 1 the year in which ARRIVE was published.

*3) Please explain why text mined articles were restricted to those with "Mice/Mouse" in the title. This eliminates many research articles. (For example, Nature is included in Figure 1—figure supplement 1, but Science is not: I suspect this is because of editorial differences in title rules, which may bias conclusions from this study). There is probably a compelling reason (e.g. automated sex and age detection breaks down when humans and mice data are described in the same article) but please tell the reader why, and note this as a limitation in the discussion.*

We have taken on board your comment and amended the Discussion and Methods sections respectively. The terms Mice/Mouse were used in the title prevented articles that made only passing references to mouse work from entering the dataset and ensured a high quality corpus for analysis. On the other hand, the PMC dataset we have used (Feb 2015 release) indeed did not have any Science papers. Although their articles are PMC Open Access, Science is not included in the PMC journal list. We note in the Open Access Subset website that “Many of the journals that deposit their complete contents into PMC include some or all of their articles in the Open Access subset” (http://www.ncbi.nlm.nih.gov/pmc/tools/openftlist/). Science has released only one issue from 2015 as open access.

4) In the Discussion, conjecture on why there is gender bias for females in many mouse studies. There are a number of references on mouse housing density – is gender assessed in these (e.g. some strains of male mice fight each other when caged, necessitating separate cages and driving up costs)?

We have addressed this point in the third paragraph of the Discussion section.